# Morphology-Tailored Gold Nanoraspberries Based on Seed-Mediated Space-Confined Self-Assembly

**DOI:** 10.3390/nano9091202

**Published:** 2019-08-27

**Authors:** Yan Yu, Yujun Xie, Pan Zeng, Dai Zhang, Rongqing Liang, Wenxing Wang, Qiongrong Ou, Shuyu Zhang

**Affiliations:** 1Department of Light Sources and Illuminating Engineering, Engineering Research Center of Advanced Lighting Technology (MoE), and Academy for Engineering and Technology, Fudan University, Shanghai 200433, China; 2Department of Chemistry and Laboratory of Advanced Materials, Fudan University, Shanghai 200433, China

**Keywords:** gold nanoraspberries (AuNRbs), space-confined effect, tunable plasmon resonances, surface-enhanced Raman scattering (SERS)

## Abstract

Raspberry-like structure, providing a high degree of symmetry and strong interparticle coupling, has received extensive attention from the community of functional material synthesis. Such structure constructed in the nanoscale using gold nanoparticles has broad applicability due to its tunable collective plasmon resonances, while the synthetic process with precise control of the morphology is critical in realizing its target functions. Here, we demonstrate a synthetic strategy of seed-mediated space-confined self-assembly using the virus-like silica (V-SiO_2_) nanoparticles as the templates, which can yield gold nanoraspberries (AuNRbs) with uniform size and controllable morphology. The spikes on V-SiO_2_ templates serve dual functions of providing more growth sites for gold nanoseeds and activating the space-confined effect for gold nanoparticles. AuNRbs with wide-range tunability of plasmon resonances from the visible to near infrared (NIR) region have been successfully synthesized, and how their geometric configurations affect their optical properties is thoroughly discussed. The close-packed AuNRbs have also demonstrated huge potential in Raman sensing due to their abundant “built-in” hotspots. This strategy offers a new route towards synthesizing high-quality AuNRbs with the capability of engineering the morphology to achieve target functions, which is highly desirable for a large number of applications.

## 1. Introduction

The raspberry, a typical fruit composed of aggregates of drupelets, has inspired the manufacture of a decent amount of micro- and nanomaterials. Such raspberry-inspired materials have been applied in a variety of fields including hydrophobic and hydrophilic coatings [1,2,3,4], micromotors [5], hierarchical templates [6,7], catalysis [8] and biomedicine [9,10]. Besides, the construction of metallic architectures, which can be achieved with raspberry-like structure as well, has become a research hotspot recently due to their superiority in providing high accessibility to catalytically active metal surfaces, which is of benefit for promising applications such as catalysis, energy storage and sensors [11,12,13,14,15]. In particular, when the raspberry-like structure is decorated with gold nanoparticles which are high-quality building blocks for complex plasmonic architectures, the collective plasmon resonances occur, which can behave completely differently from the individual ones [16,17,18,19,20]. The high degree of symmetry and strong interparticle coupling of the raspberry-like structure offers a new platform to tailor the plasmonic and magnetic resonances of nanoparticles [21,22,23,24,25]. To achieve better performance, the precise control over the well-defined morphology and interparticle distance is critical [25,26], indicating that the fabrication process plays an essential role in maximizing the function of raspberry-like gold nanoparticles.

Such raspberry-like nanostructures can be fabricated by bottom-up strategies, which favor the fabrication of three-dimensional clusters [27,28]. Pre-formed gold nanoparticles were self-assembled to form gold nanoraspberries (AuNRbs) via electrostatic interaction [21,24,29], covalent bonding [21] and bio-mediated attraction [30,31]. The nonuniformity of interparticle gaps is the main limitation of these approaches since the gold nanoparticles are randomly distributed on the core [24]. To further reduce the interparticle distance, Fakhraai and Park’s group came up with the templated surfactant-assisted seed-growth method, allowing for in-situ growth and assembly of gold nanobeads around a polystyrene core, which then yielded a raspberry-like structure with close-packed gold nanobeads [22,23,25]. The surfactant acts as a stable colloid which is indispensable in the growth process, however, it covers the clean surface of plasmonic nanoparticles, so that the accessibility of the hotspots by target analytes or building nanoblocks/nanosheets could be compromised [32].

In this work, we report a synthetic strategy of seed-mediated space-confined self-assembly to construct well-defined AuNRbs. The recently reported virus-like silica (V-SiO_2_) nanoparticles [33] are used as the templates with dual functions, the spiky surfaces of which generate more active sites for gold nanoseeds to assemble and meanwhile confine the growth space of gold nanoparticles to form high-quality AuNRbs. This method opens up a new dimension of engineering the morphology of AuNRbs and it works without the aid of surfactants. The obtained AuNRbs exhibit structural coloration with wide-range tunability due to the localized resonances of plasmonic clusters. The closely packed structure with significantly reduced interparticle distance substantially boosts the signals when the nanoraspberries are used as surface enhanced Raman scattering (SERS) substrates. The theoretical analysis of the electric field provides physical insights that can guide the synthesis of AuNRbs with designed properties for broad applicability.

## 2. Materials and Methods

### 2.1. Chemicals and Materials

All chemicals were purchased from commercial suppliers and used without further purification. Cetyltrimethyl ammonium bromide (CTAB, >98.0%), Tetrakishydroxymethylphosphonium chloride (THPC, 80% in H_2_O) and hydroxylamine hydrochloride (99.995%) were purchased from Sigma-Aldrich (Shanghai, China). Gold chloride trihydrate (HAuCl_4_·3H_2_O, ≥99.9%), ammonium hydroxide solution (28% in H_2_O), tetraethyl orthosilicate (TEOS), (3-Aminopropyl) triethoxysilane (APTES, 98%), sodium hydroxide (NaOH, ≥98%), cyclohexane (≥99.9%) and potassium carbonate (K_2_CO_3_, 99%) were purchased from Aladdin (Shanghai, China)

### 2.2. Preparation of Positively Charged V-SiO_2_ and S-SiO_2_ Nanoparticles

The V-SiO_2_ nanoparticles were synthesized via an epitaxial growth approach reported by Wang et al. [33]. Briefly, 20 mL TEOS in cyclohexane (20 *v/v*%) was added to mixed aqueous solution containing 1.0 g CTAB and 0.8 mL NaOH (0.1 M) with a total volume of 50 mL and was stirred for 48 h at 60 °C. The obtained solution of V-SiO_2_ nanoparticles was centrifuged at 12,000 rpm for 8 min and washed by water and ethanol repeatedly under the same centrifugation conditions. After being calcinated at 400 °C for 4 h to remove CTAB templates, the powdered sample was collected. S-SiO_2_ nanoparticles were prepared by a modified Stöber method. In detail, 6 mL of TEOS was added to a mixed solution containing 75 mL ethanol, 10 mL deionized water and 1.6 mL ammonia aqueous and kept stirring overnight. The products were washed by water and ethanol for several times through centrifugation, and then collected after drying at 120 °C. For the amino modification, 50 mg samples were dispersed in 30 mL toluene containing 300 μL APTES and refluxed for 8 h at 110 °C. The positively charged V-SiO_2_ and S-SiO_2_ samples were then obtained and dispersed in deionized water after washing by water for several times.

### 2.3. Preparation of Gold Nanoseeds

The gold nanoseeds with a diameter of ~2 nm were prepared via a THPC reduction process [34]. Typically, 1 mL THPC diluted solution (1.2 mL 80% aqueous solution diluted to 100 mL with water) was added to a 47 mL aqueous solution containing 3 mL NaOH (0.1 M) and stirred for 2 min. Then 0.5 mL HAuCl_4_ (100 mM) was added into the mixed solution under vigorous stirring and the color of the solution changed to brown simultaneously, which indicates the formation of gold nanoseeds.

### 2.4. Preparation of Gold Nanocomposites (AuNCs) and AuNRbs

For the preparation of AuNCs, the aqueous solution V-SiO_2_ nanoparticles and gold nanoseeds were mixed and stirred for 4 h. The resulting mixture was repeatedly centrifuged with water at 10,000 rpm for 8 min, and the precipitate was collected and redispersed in deionized water with a resulting concentration of ~10 mg/mL. The brown color of the precipitates indicates the successful attachment of gold nanoseeds onto the surface of V-SiO_2_ nanoparticles and the dispersion of ~10 mg/mL has a golden color. For the preparation of AuNRbs, 200 μL AuNC solution was added into a growth solution containing 10 mg K_2_CO_3_, a pre-setting amount of HAuCl_4_ and 40 mL deionized water. Finally, 10 μL hydroxylamine hydrochloride (250 mg/mL) was injected into the mixture as the reducing agent during vigorous stirring and the raspberry-like structures were subsequently formed. As the concentration of HAuCl_4_ in the growth solution increased, the morphology of the nanoraspberries gradually plumped up. The relationship between the morphology of AuNRbs and the concentration of HAuCl_4_ in the growth solution will be discussed in detail later.

### 2.5. Preparation of AuNP-20

The gold nanoparticles were prepared via a slightly modified seed growth method used in our prior work [20]. Briefly, 50 mL HAuCl_4_ solution (0.25 mM) was heated to boiling and then 1.5 mL of 34 mM sodium citrate solution was rapidly injected into the boiling solution and kept boiling for 10 min. After the solution was cooled down to 80 °C, 20 mL reaction mixture was extracted. Then, 17 mL of deionized water, 0.2 mL of 0.25 mM HAuCl_4_ solution and 2.8 mL of 34 mM sodium citrate solution were added subsequently into the extracted solution. The system was kept stirring at 80 °C for 10 min and the citrate-stabilized AuNP-20 was obtained.

### 2.6. Sample Characterization

Transmission electron microscopy (TEM) images were acquired on a JEM-1400 microscope operating at 120 kV. High-resolution TEM (HRTEM) imaging were performed on JEM-2100F microscope (JEOL, Beijing, China) with an accelerating voltage of 200 kV equipped with a post-column Gatan imaging filter. Extinction spectra were recorded using a Cary 5000 UV/Vis/NIR spectrophotometer. Scanning electron microscopy (SEM) imaging was performed with a field emission SEM (Sigma VP, Carl Zeiss AG, Jena, Germany) at 3 kV utilizing the in-lens detector at a working distance of approximately 3 mm.

### 2.7. Raman Measurements

The method of drop-deposited Raman spectroscopy was utilized for surface enhanced Raman scattering (SERS) measurement [35,36]. The solutions containing AuNCs, AuNRb-15, AuNRb-22 or AuNP-20 were concentrated by centrifugation and mixed with R6G solution. The final concentration of R6G was 10^−5^ M. The samples were allowed to stand for 1h after shaking. Then, 10 µL of the sample was dropped and dried on a silicon wafer and ready for Raman tests. Raman spectra were measured by a Renishaw inVia Raman microscope system equipped with He–Ne laser with an excitation wavelength at 633 nm. The laser was focused on the samples through a 50× objective (Leica, numerical aperture: 0.5). The Raman tests were conducted with a laser power of 0.85 mW and an acquisition time of 10 s.

### 2.8. COMSOL Simulations

The optical properties of gold nanoraspberries were calculated using COMSOL Multiphysics. By resembling the TEM results of AuNRbs in different growth stages, the geometric configurations of gold nanoraspberries were defined for the calculation. In the geometric configurations, the silica spikes were not included, since they have negligible influence on the absorption and scattering resonance. The gold nanoparticles were assumed to be nanospheres with the same diameters for computational convenience. Therefore, a gold nanoraspberry in the model consisted of a dielectric core with a diameter of 150 nm and multiple identical gold nanospheres decorated on the surface of the core. Geometric symmetry was employed, which assumes a uniform distribution of gold nanospheres on the surface. According to the experimental results, the nanosphere diameter of AuNRb-8, AuNRb-11, AuNRb-15 and AuNRb-22 was set to be 8.3 nm, 11.4 nm, 15 nm and 22 nm, respectively. The interparticle distance (surface to surface) of AuNRb-8, AuNRb-11, AuNRb-15 and AuNRb-22 was set to be 4.1 nm, 3.9 nm, 3.5 nm and 0.5 nm, respectively. The model assumes the nanoraspberries are isolated, so no interaction between different nanoraspberries is taken into account. Limited by the computational capacity of our workstation, we built a slice plane passing through the center of the dielectric sphere for calculation, with both the propagation direction of the incident light and the electric field polarization sitting in the plane. The absorption cross-sections were obtained by integrating the total power dissipation density over the volumes of all the gold nanospheres. The scattering cross-sections were obtained by integrating the relative normal Poynting vector (which is defined from the outwards-facing normal vector and the automatically defined coordinate components of the Poynting flux) over the surfaces of all the gold nanospheres. The extinction cross-section was the sum of the absorption cross-section and the scattering cross-section. The parameter sweep of wavelength ranged from 400 nm to 1000 nm with an interval of 5 nm. The dielectric functions for gold were based on Johnson and Christy’s calculations [37].

## 3. Results

The schematic diagram of the fabrication process of AuNRbs is shown in Figure 1a. The amino functionalized V-SiO_2_ nanoparticles with spiky surface were first prepared, which had an average diameter of ~150 nm with good uniformity and dispersity. A TEM image of the V-SiO_2_ nanoparticles is presented in Figure 1b and the spikes were estimated to be ~10 nm in length and ~6 nm in diameter with a density of about 0.0023/nm^2^ according to the TEM image. The gold nanoseeds with an average diameter of ~2 nm were then assembled on the surface of V-SiO_2_ nanoparticles by electrostatic force. After the electrostatic adsorption, a large number of gold nanoseeds were uniformly distributed on the rough surface of V-SiO_2_ nanoparticles, forming gold nanocomposites (AuNCs) as shown in Figure 1c. By adding the growth solution, the gold nanoseeds were transformed into nanoparticles, and the raspberry-like structure was subsequently formed. Figure 1d,e show the TEM image and SEM image of the AuNRbs, respectively, both of which demonstrate the high-quality and uniformity of the AuNRbs. Figure 1f,g present a direct comparison of the closeup photograph of a single raspberry fruit and the TEM image of a single AuNRb. The high degree of morphological similarity suggests the success of this synthetic approach.

The growth process of AuNRbs was systematically investigated and a close correlation between the structure and the corresponding optical property was revealed. AuNRbs at different growth stages were synthesized by controlling the amount of HAuCl_4_ in the growth solution (from 0.025 mM to 0.225 mM). The TEM image of AuNCs and those of AuNRbs at different growth stages are shown in Figure 2a–e. Good uniformity and dispersity of AuNRbs were maintained throughout the growth process. With the increasing amount of HAuCl_4_, the size of gold nanoparticles increased and the interparticle distances reduced, forming a close-packed raspberry-like structure. The size distributions of the assembled gold nanoparticles during the growth process were calculated according to the TEM images and the results are summarized in Figure 2f–j. In detail, three TEM images were selected for every sample, each image measured the sizes of around forty gold nanoparticles assembled on the AuNRb. The scatter plots of around 120 points for each sample are shown in Appendix A. The size of the gold nanoseeds in AuNC (Figure 2a) were estimated to be around 2 nm in diameter, and that of gold nanoparticles at different growth stages of AuNRbs (Figure 2b–e) ranged from 8.3 nm to 22.0 nm. In accordance with the sizes of assembled gold nanoparticles, the AuNRbs at different growth stages were termed AuNRb-8, AuNRb-11, AuNRb-15 and AuNRb-22. The AuNRbs exhibited great stability. Taking AuNRb-15 as an example, the morphology remained the same after 4 months storage as indicated by the TEM images shown in Appendix A. Interestingly, the collective optical properties of the colloids constantly varied during the growth process. The color of the colloids changed from golden to red, purple, blue and cyan as the gold nanoparticles gradually grew (the insets in Figure 2k–o), exhibiting the capability of plasmonic color generation with wide-range tunability. The corresponding extinction spectra during different growth stages are shown in Figure 2k–o. No localized surface plasmon resonance (LSPR) band was observed for AuNCs in the range from 300 nm to 1200 nm, however, a characteristic LSPR peak appears during the growth process. Accompanying with the continual growth, the LSPR peak was broadened and red-shifted from visible to NIR region, which is attributed to the growing size of assembled gold nanoparticles and the enhanced interparticle coupling with a reduced distance [38,39]. Moreover, the size distribution became more discrete as the size of gold nanoparticles increases (as indicated by Figure 2f–j), which also leads to a spectrum broadening [23].

The optical properties of AuNRbs were calculated using COMSOL Multiphysics, which accounts for the variations of electric field within and between the gold nanoparticles based on a finite element method. The geometric configurations for the calculation resemble approximately the TEM results shown in Figure 2. Since the size of the assembled gold nanoparticles and the SiO_2_ template has been determined according to the TEM images, the number of gold nanoparticles was associated with the interparticle distance when the gold nanoparticles were uniformly distributed on the SiO_2_ template. As the range of the interparticle distance can be also obtained from the TEM images, a reasonable integral number can be determined by fixing the right distance value within the range. Finally, the model for calculation has been established and a detailed description of parameters and settings can be found in the Section 2. The extinction cross-sections of AuNRb-8, AuNRb-11, AuNRb-15 and AuNRb-22 were obtained and the results are presented in Figure 3a. Although the calculations were derived from a number of simplified assumptions, the results were reasonably consistent with the experimental ones. The extinction cross-section was composed of the absorption cross-section and scattering cross-section. The increasing size of gold nanoparticles and the decreasing interparticle distance led to the increasing intensities and spectral broadening of both cross-sections and a red-shift of the absorption and scattering resonance peaks. In order to investigate how the nanoparticle size and interparticle distance independently affect the extinction cross-section, we calculated the cross-section as a function of size and a function of distance, respectively, and the results are shown in Figure 3b,c. When the interparticle distance was kept at 4 nm and the nanoparticle size increased from 6 nm to 16 nm in diameter (Figure 3b), the position of absorption peak was barely shifted with a slight increase in the peak intensity which was due to the increasing volume of gold nanoparticles. On the contrary, the peak position of scattering resonance was red-shifted with an intensity enhancement and peak broadening. As the size increases, an additional scattering peak emerges at the shorter wavelength region, which supports a different resonance coupling (Appendix A). When the diameter of the nanoparticle was kept at 10 nm and the interparticle distance was reduced from 5.2 nm to 0.9 nm, similar behavior of the scattering resonance was observed, however, the absorption peak showed a distinct red-shift which was caused by the circumferential absorption enhancement due to the interparticle resonance (Appendix A). So, the results indicate the interparticle distance had a more distinct effect on the absorption of gold nanoraspberries, and both the distance and nanoparticle size were capable of significantly affecting the scattering resonance.

Based on the observation of the growth process, we explored the formation mechanism of this delicate raspberry-like structure and illustrate it in Figure 4a. Firstly, the gold nanoseeds were attached onto the surface of V-SiO_2_ nanoparticles via electrostatic force and AuNCs were subsequently formed. In this process, the surfaces of the spikes were extensively covered by gold nanoseeds since the size of the spikes (~10 nm in length and ~6 nm in diameter) was larger than that of the gold nanoseeds (~2 nm in diameter). The red ellipse marked in Figure 4b shows the evidence of a representative spike covered with plenty of gold nanoseeds on the surface of a single AuNC. To investigate the impact of the spikes, silica nanoparticles with smooth surfaces (S-SiO_2_), which have been widely used as templates, were also prepared (Appendix A). Note that the S-SiO_2_ nanoparticles used for comparison had the same diameter as the V-SiO_2_ in order to eliminate the influence of relative size or curvature. Apparently, provided with more growth sites, the amount of gold nanoseeds attached on V-SiO_2_ nanoparticles (Figure 4b) far exceeded that on S-SiO_2_ nanoparticles (Appendix A), thus enabling uniform growth across the entire surface to form a complete structure at the next stage. The growth process subsequently kicked off when the growth solution was added. The aggregative growth dominated this process and formed size-increasing gold nanoparticles [40,41], since Au^3+^ ions were reduced on the surfaces of gold nanoseeds which had already been anchored on the surface of the silica core. As the gold nanoparticles grew larger, the adjacent nanoparticles started to contact each other and form agglomerations. Generally, if the aggregative growth was conducted perfectly on an S-SiO_2_ nanoparticle, a continuous gold nanoshell could be obtained, which was experimentally verified and presented in the Appendix A. In contrast to S-SiO_2_ nanoparticles, the spiky surfaces of V-SiO_2_ nanoparticles caused a space-confined effect during the aggregative growth. Instead of forming a continuous nanoshell, the spikes limited the size of agglomerations, leading to the formation of the raspberry-like structure. Figure 4c,d shows the TEM images of AuNRbs at different growth stages. The blue dotted boxes in the two figures show the gold nanoparticle-surrounded silica spikes which play a critical role in the space-confined effect. For better comparison, S-SiO_2_ and V-SiO_2_ silica templates were removed by hydrofluoric acid etching and the resulting products are shown in Appendix A. The product using a V-SiO_2_ template had a clear raspberry-like structure with an undulating outside surface, while the product from an S-SiO_2_ template had an incomplete shell with a much smoother outer surface. The incompleteness of the gold nanoshell was caused by the insufficient growth sites due to the sparsely distributed gold nanoseeds on the S-SiO_2_ template. Therefore, the spikes on V-SiO_2_ templates serve dual functions of providing more growth sites and activating the space-confined effect.

It is well known that closely packed gold nanoparticles can give rise to extraordinarily enhanced Raman signals [42,43,44], therefore, AuNRbs are excellent candidates to generate strong SERS signals. The drop-deposited method was utilized for SERS measurement [35,36] and the schematic diagram was shown in Figure 5a. Figure 5a shows the schematic diagram of a SERS measurement. The SERS performances of AuNCs, AuNRb-15 and AuNRb-22 were characterized using Rhodamine 6G (R6G) as the probe molecules with a concentration of 10^−5^ M and the results are shown in Figure 5b. AuNCs were unable to generate detectable SERS signals of R6G due to the small size (~2 nm) and weak coupling of the gold nanoseeds. This was consistent with the result that no LSPR peak could be found in its extinction spectrum (Figure 2k). For AuNRb-15 and AuNRb-22, well-resolved Raman vibrational signals of R6G were observed. AuNRb-22 could significantly amplify the SERS signal due to the LSPR of gold nanoparticles with increased size and the boosted hotspots attributed to small interparticle nanogaps. In addition, the SERS performance of gold nanoparticles with a diameter of ~20 nm (denoted as AuNP-20) was measured under the same conditions for comparison with AuNRb-22, since the diameter of AuNP-20 was close to that of the assembled gold nanoparticles of AuNRb-22. For the characteristic peak of R6G at 1508 cm^−1^, the SERS signal generated from AuNRb-22 was estimated to be around 30,000 counts, which was more than an order of magnitude stronger than that from AuNP-20, whose SERS intensity was estimated to be around 2000 counts. It is therefore obvious that the AuNRbs formed by close-packed gold nanoparticles had superior SERS performance compared with the random aggregations of gold nanoparticles with similar size. Besides, the SERS performance of the sample fabricating from S-SiO_2_ templates through the same growth procedure as that of the AuNRb-22 (denoted as S-SiO_2_-AuNPs) was measured for comparison as well. The results showed that the SERS signal of AuNRb-22 was approximately six times that of S-SiO_2_-AuNPs at 1508 cm^−1^. The better performance of AuNRb-22 was attributed to the abundant hotspots provided by the closely-packed raspberry-structure. Besides, the AuNRbs was reported to be of great significance for designing reproducible SERS substrates [45,46], since the abundant “built-in” hotspots can lead to extraordinary weak distance dependence in SERS [22]. The results of our SERS measurement not only demonstrate the engineerable interparticle distance in this raspberry-like structure, but also indicate a huge potential of AuNRbs in SERS applications.

## 4. Conclusions

We demonstrate a synthetic strategy of seed-mediated space-confined self-assembly to fabricate delicate AuNRbs with uniform size and controllable morphology. The spikes of V-SiO_2_ that were used as templates serve dual functions, which not only provide more growth sites for gold nanoseeds, but also confine the growth space of gold nanoparticles to form a high-quality raspberry-like structure. The synthesized AuNRbs support LSPR with wide-range tunability and thus engineerable structural coloration. The effects of nanoparticle size and interparticle distance on the optical properties of AuNRbs have been thoroughly investigated. Moreover, the close-packed AuNRbs demonstrate huge potential in Raman sensing due to their abundant “built-in” hotspots. This method offers a feasible and generalized route towards the synthesis of metal nanoclusters with well-controlled morphology. By adjusting the size of the assembled metal nanoparticles and the gaps inside the raspberry-like structure, the active sites and the accessibility to active surface can be easily tuned, thus benefiting diverse applications including magnetic resonance, SERS and catalysis.

## Figures and Tables

**Figure 1 nanomaterials-09-01202-f001:**
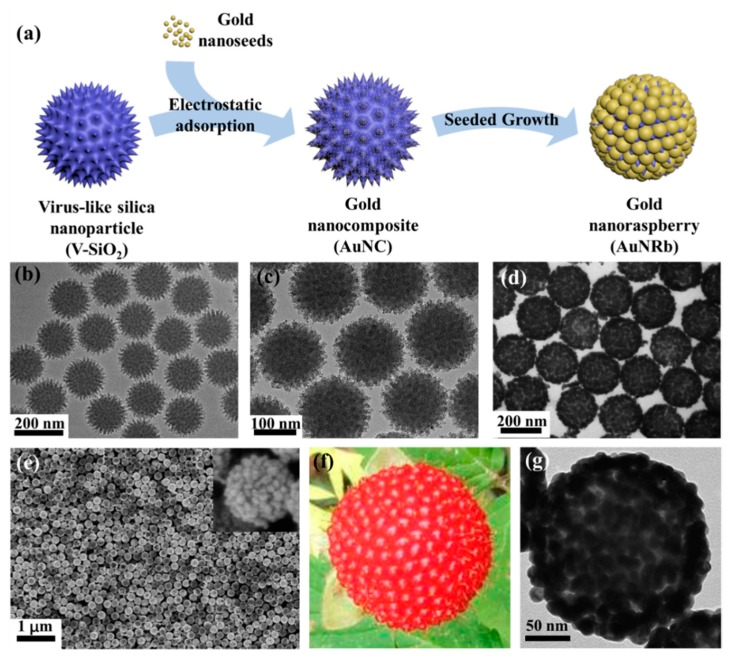
(**a**) A schematic diagram of the fabrication process of gold nanoraspberries (AuNRbs). Transmission electron microscopy (TEM) images of (**b**) virus-like silica (V-SiO_2_) nanoparticles; (**c**) gold nanocomposites (AuNCs) formed by gold nanoseeds-attached V-SiO_2_ nanoparticles; and (**d**) AuNRbs after the growth process. (**e**) The scanning electron microscopy (SEM) image of AuNRbs. The inset figure shows a single AuNRb. (**f**) A closeup photograph of a single raspberry fruit. (**g**) The TEM image of a single AuNRb.

**Figure 2 nanomaterials-09-01202-f002:**
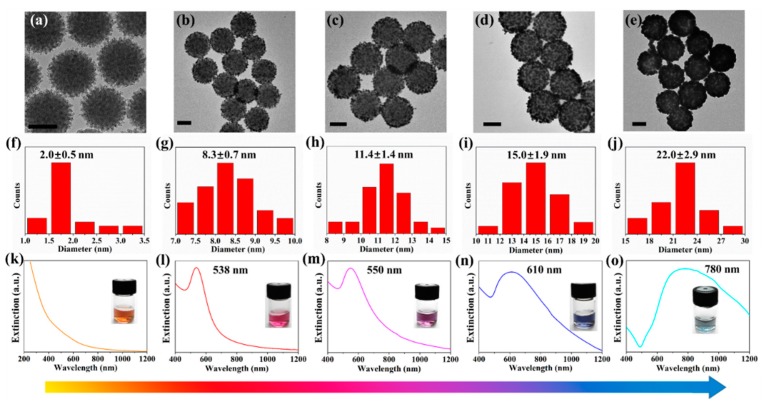
A TEM image of (**a**) AuNCs, and AuNRbs synthesized using a HAuCl_4_ solution with a concentration of (**b**) 0.025 mM, (**c**) 0.0375 mM, (**d**) 0.075 mM and (**e**) 0.225 mM, respectively. The scale bar is 100 nm. (**f**–**j**) The size distribution of assembled gold nanoseeds and gold nanoparticles corresponding to (**a**–**e**), respectively. (**k**–**o**) The extinction spectrum with an inset showing the photo of colloid solution corresponding to (**a**–**e**), respectively.

**Figure 3 nanomaterials-09-01202-f003:**
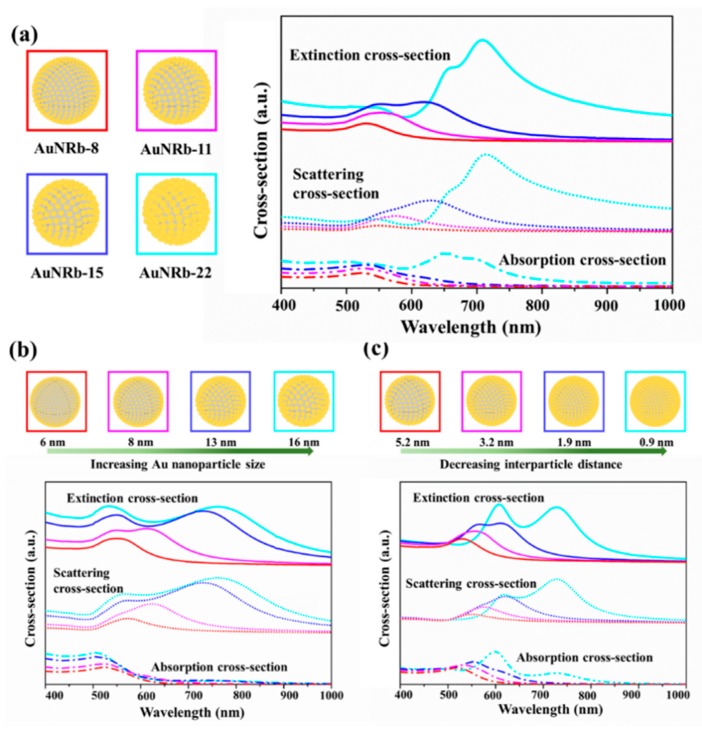
(**a**) The calculated absorption cross-sections, scattering cross-sections and extinction cross-sections of AuNRb-8 (red), AuNRb-11 (magenta), AuNRb-15 (blue) and AuNRb-22 (cyan). The extinction cross-section is the sum of the absorption cross-section and the scattering cross-section. (**b**) The dependence of cross-sections as a function of the nanoparticle size when the interparticle distance is kept at 4 nm. The sizes of the nanoparticles used in simulation increased from 6 nm (red) to 8 nm (magenta), 13 nm (blue) and 16 nm (cyan). (**c**) The dependence of cross-sections as a function of the interparticle distance when the nanoparticle size is kept at 10 nm. The interparticle distances of the nanoparticles used in simulation decreased from 5.2 nm (red) to 3.2 nm (magenta), 1.9 nm (blue) and 0.9 nm (cyan). The models used in simulation are displayed in boxes with border colors representing different conditions.

**Figure 4 nanomaterials-09-01202-f004:**
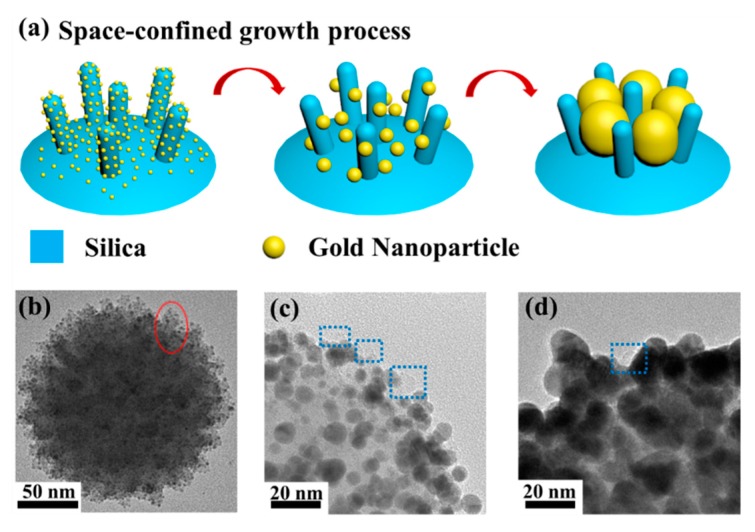
(**a**) A schematic diagram illustrating the formation mechanism of the AuNRb structure. (**b**) The TEM image of a single AuNC. The red ellipse highlights a spike covered with gold nanoseeds. (**c**) The close-up TEM image of a AuNRb-8. Inside the dotted blue boxes are the silica spikes. (**d**) The close-up TEM image of a AuNRb-15. Inside the dotted blue box is a silica spike.

**Figure 5 nanomaterials-09-01202-f005:**
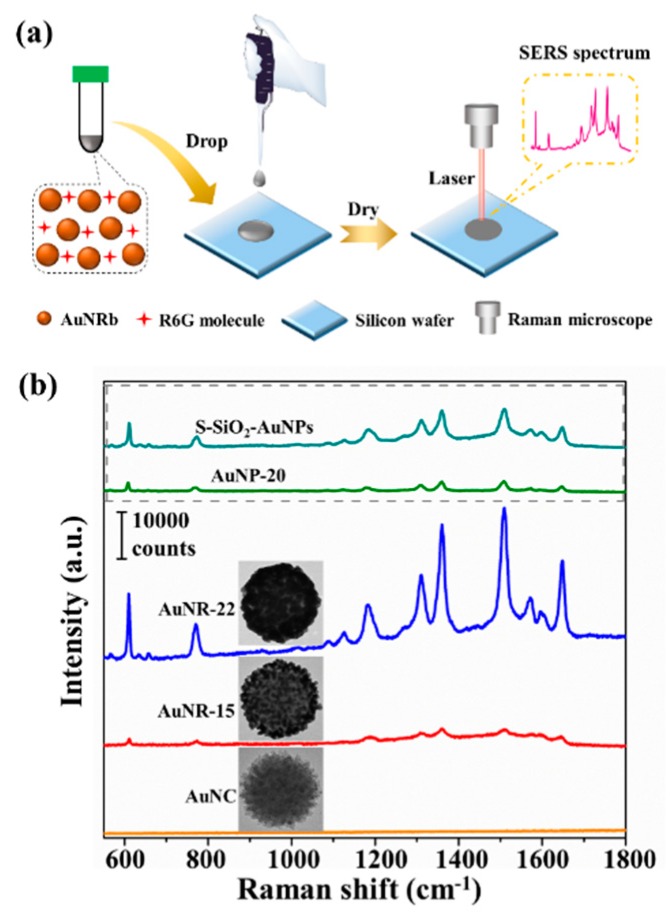
(**a**) A schematic diagram of surface enhanced Raman scattering (SERS) detection. (**b**) The SERS performance of AuNC, AuNRb-15, AuNRb-22 and AuNP-20 using Rhodamine 6G (R6G) as the probe molecules with a concentration of 10^−5^ M excited by a 633 nm laser.

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
