# Peer review of "Morphology-Tailored Gold Nanoraspberries Based on Seed-Mediated Space-Confined Self-Assembly"

_nanomaterials, 2019, doi:10.3390/nano9091202_

Round 1
Reviewer 1 Report
The article submitted by Yu et al. concerns the design of nanoparticles of raspberry morphology and their manufacture according to a recipe that would imitate the granular and nevertheless regular structure of the real fruit. The authors propose for this purpose to use silica substrates bearing spikes that serve as templates to control the morphology and size of gold grains at the time of their seeded regrowth. The study also claims potential applications in SERS.
Although very well written and illustrated, the article unfortunately suffers from several weaknesses.
In the first place, there are errors or misunderstandings in principle. For example, on page 5 line 192, the authors consider that the observed color changes are related to structural coloration which is unrelated to the use of plasmonic particles.
Then the methodology is unclear, especially how the size histograms (Figure 2) were established. If it is on the basis of the TEM images that are just above, then the uncertainty on the measurements must be enormous and consequently on the polydispersity in size as well. Also, the modelizations of figure 3 were not systematically made on the same sizes of nanoparticles as the materials.
Although it can not be disputed, the SERS effect is low compared to the performance reported recently.
Finally, the model of growth between spikes proposed in Figure 4a is rather naive and one could just as easily imagine that the effect of the spikes is just to create possibly evenly spaced holes in the gold coating (compared to the same coating obtained on smooth substrates).
Since the gold seeded regrowth recipe on silica is not original, the SERS performance is average and the growth model is not well founded, I think this work does not deserve to be published in Nanomaterials journal.
Reviewer 2 Report
The authors demonstrate a synthetic strategy of seed-mediated space-confined self-assembly using the virus-like silica (V-SiO2) nanoparticles as the templates, which can yield gold nanoraspberries (AuNRbs) with uniform size and controllable morphology. This work collects lots of data on the target subjects, but many exciting aspects are well included. The authors show many characterization data to support the claims. The manuscript is well organized. After the following minor revision, I can accept the publication of this paper. How about the material stability? It is better to add more comments on this. The authors show wide-angle XRD patterns for samples. How about the average crystallite sizes? This size is matched with TEM data? Related papers have been reported by different research groups. It is better to cite the following refs to support some related paragraphs in the introduction part. [Review] Nano Today, Volume 21, August 2018, Pages 91-105 Acc. Chem. Res., 2018, 51 (8), pp 1764–1773 Chem. Sci., 2019, 10, 4054-4061 Small 9 (7), 1047-1051, 2013 Chemical Communications 46 (21), 3684, 2010 Overall the manuscript is well written, but I want to see the authors' perspective on this research in the conclusion part.
Round 2
Reviewer 1 Report
The main explanations and alterations to the first version are satisfactory.
Reviewer 2 Report
Highly improved